# MousePZT: A simple, reliable, low-cost device for vital sign monitoring and respiratory gating in mice under anesthesia

**Daniel A. Rivera**[1], **Anne E. Buglione**[1,2], **Sadie E. Ray**[2], **Chris B. Schaffer**[1] *

**1** Nancy E. and Peter C. Meinig School of Biomedical Engineering, Cornell University, Ithaca, New York, United States of America, **2** College of Veterinary Medicine, Cornell University, Ithaca, New York, United States of America

* cs385@cornell.edu

## Abstract

Small animal studies in biomedical research often require anesthesia to reduce pain or stress experienced by research animals and to minimize motion artifact during imaging or other measurements. Anesthetized animals must be closely monitored for the safety of the animals and to prevent unintended effects of altered physiology on experimental outcomes. Many currently available monitoring devices are expensive, invasive, or interfere with experimental design. Here, we present MousePZT, a low-cost device based on a simple piezo-electric sensor, with a custom circuit and computer software that allows for measurements of both respiratory rate and heart rate in a non-invasive, minimal contact manner. We find the accuracy of the MousePZT device in measuring respiratory and heart rate matches those of commercial systems. Using the widely-used gas isoflurane and injectable ketamine/xylazine combination, we also demonstrate that changes in respiratory rate are more easily detected and can precede changes in heart rate associated with variations in anesthetic depth. Additional circuitry on the device outputs a respiration-locked trigger signal for respiratory-gating of imaging or other data acquisition and has high sensitivity and specificity for detecting respiratory cycles. We provide detailed instruction documents and all necessary microcontroller and computer software, enabling straightforward construction and utilization of this device.

## Introduction

The use of anesthetics in mice and other small laboratory animals is essential for many experimental procedures, but comes with risks, even in healthy animals. Isoflurane is an inhalant anesthetic commonly used due to its rapid-onset, short recovery time, and the ease of adjusting the anesthetic depth [1, 2]. The effective anesthetic dose depends on a variety of animal-dependent factors, so that fixing the percentage of isoflurane in the inhaled gas mixture can lead to over- and under-dosing of some animals. Insufficient or 'light' anesthesia risks awareness during stressful or painful procedures, while excessive or 'deep' anesthesia can lead to dangerous

**Data Availability Statement:** All resources and underlying data are available at https://github.com/sn-lab/Mouse-Breathing-Sensor.

**Funding:** "This work was supported by the National Institute on Aging (https://www.nia.nih.gov/), grant number, AG049952 (CBS). The funders had no role in study design, data collection and analysis, decision to publish, or preparation of the manuscript. There was no additional external funding received for this study".

**Competing interests:** The authors have declared that no competing interests exist.

decreases in heart rate, respiratory rate, and blood pressure [1, 2]. In addition, anesthetic agents alter a range of physiological parameters that can impact experimental results if not well-controlled across animals in an experiment. For example, in neurological and neurovascular research, isoflurane has been implicated in burst suppression [3–5], decreased cerebral metabolism [3, 5, 6], increased vasodilation [3, 4, 6], suppressed vasomotion [6, 7], altered neurovascular coupling responses [3, 4, 6], and neuroprotection against ischemia [3]. Minimizing the variability in experiments due to these effects requires a narrowly restricted range of anesthetic depth, if anesthesia cannot be avoided. In addition, volatile anesthetics like isoflurane can accumulate in the body due to their fat solubility [8], necessitating lowered inhaled isoflurane doses over time to maintain the same level of anesthetic depth [9]. Monitoring heart and respiratory rates, which depend sensitively on anesthetic depth, allows for adjustments of the anesthetic dose to ensure appropriate anesthetic depth, detect side effects of administered drugs, and identify acute adverse events and trends during any procedure.

Several tools and techniques are available to monitor vital signs, including pulse-oximeters, electrocardiograms (ECG), blood pressure cuffs, arterial catheters, capnographs, and thermometers [10]. Labs may abstain from using some of these tools due to their invasiveness, the complexity of incorporating the tools into an experiment, poor reliability under anesthesia, or the overall cost. For example, arterial catheters provide excellent measures of blood pressure but are very invasive, requiring surgical implantation with risk of complications during implantation. Prolonged monitoring additionally comes with the potential for dampened signals requiring the use of heparin or flushing, which is further limited by the small blood volume in mice [11, 12]. Rodent blood pressure cuffs placed around the tail measure blood pressure in the tail artery, but require warming the tail and often underestimate blood pressure, especially in anesthetized animals [13]. ECG requires at least three leads to be attached to the limbs by subcutaneous needles, direct superficial contact, either by placing the animal on a conductive platform, or positioning leads on the skin. Subcutaneous needles cause tissue damage and pain upon recovery from the procedure. Although clips can be modified to reduce skin damage and conductive platforms are atraumatic, the additional wires and restricted positioning on the platforms can quickly complicate the experimental setup. Further, the ECG signal is prone to electrical interference. For pulse-oximetry, dark hair must be removed from the measurement area for clear optical access. Sensor placement needs to be adjusted periodically to ensure adequate signal amplitude, and extended use of some sensors (e.g. thigh clips) can cause underlying soft tissue injury from compression. Capnographs are simple to incorporate into exhalation lines but are more reliable with intubated or tracheotomized mice, as compared to free breathing animals. Even in appropriate experiments, they require additional calibration steps and can be costly. Surveying across all available devices, the live animal research community would benefit from simple, reliable, and low-cost systems to monitor and record vital signs of mice that can be straightforwardly incorporated into surgical and experimental procedures.

Of the vital signs often used during free-breathing anesthesia, respiration is one of the simplest to measure and provides a sensitive assay of anesthetic depth [10]. Respiratory rate and effort can be evaluated by the experimenter by visually assessing frequency and pattern of chest wall excursions. However, some experimental setups, such as *in vivo* imaging are incompatible with continuous, direct visualization of the animal. Furthermore, this approach may be error prone and forgoes a thorough log of respiratory rate. Approaches to quantify respiration include measurement of muscle activity via electromyography, face masks or temperature probes to measure respiratory gas movement, and cameras or movement sensors to detect movement of the body wall [14]. Previously developed low-contact or low-cost respiratory monitoring systems for use in research utilized changes in infrared light reflectivity [15] or the

amplitude and phase of reflected radio frequency signals [16, 17] due to chest wall motion, in temperature due to expired air [18], and, more commonly, in voltage signals from piezoelectric transducers (PZT) that were abutting the mouse [19–24]. PZTs have the advantage of being both low-cost and highly sensitive to pressure perturbations due to breathing. Few of the previously developed piezo-based sensors, however, can detect both respiratory rate and heart rate. Here, we present a simple, low-cost PZT-based system, MousePZT, that allows for measurement of both respiratory rate and heart rate with accuracy matching that of ECG and pulse-oximetry.

In addition to monitoring vital signs, many live animal imaging modalities, such as MRI, micro-CT, PET, and optical or ultrasound imaging benefit from the ability to synchronize data acquisition with the respiratory cycle of the animal to minimize motion artifact, requiring a trigger signal indicating inspiration that can gate the imaging acquisition [12]. The system has additional circuitry for real-time detection of respiration with high specificity and sensitivity, allowing respiratory gating during data acquisition in small animal imaging experiments.

## Results

The MousePZT system to measure mouse respiratory and heart rate uses a piezoelectric disk placed below the anesthetized mouse (Fig 1A). The electrical signals produced by deflections of the thorax or abdomen are amplified and filtered, and the voltage values are output by a microcontroller to a computer app (Fig 1B) for signal processing to determine the respiratory and heart rates. MousePZT also includes an analog circuit for respiratory peak detection (Fig 1C) that outputs a trigger signal (Fig 1D), which would allow for gating of data acquisition.

### Testing the accuracy of the MousePZT respiratory trigger

We tested the utility of the respiratory waveforms and the accuracy of the respiratory trigger circuit in four mice anesthetized with either isoflurane (n = 3) or a mixture of ketamine and xylazine (n = 1), two anesthetics widely used in laboratory settings. To demonstrate the sensitivity of measurement to the orientation of the mouse (dorsal, lateral, and ventral recumbency), two mice were recorded for 15–30 s under 1–2% isoflurane anesthesia (Fig 2A). The output TTL signal from the respiratory trigger circuit matched well with respiratory epochs determined by digital signal processing of the analog circuit output (Fig 2B). Averaged session-to-session sensitivity and specificity were 98.6% and 96.0%, respectively, based on the presence or absence of TTL pulses during respiratory or inter-respiratory epochs, respectively. The average respiratory rate determined by output onset times of the TTL pulses was highly correlated ($R^2$ = 0.997) with that determined from the analog circuit output (Fig 2C) using digital peak detection. We further compared the respiratory trigger with visually assessed respiration, based on excursions of the body wall (S1 Movie) from three mice to ensure the trigger appeared during the respiratory cycle. Based on visual scoring, there were no missed respirations, and all pulses occurred during respiratory motion, with short pulses occurring central to the cycle (n = 133 breaths, CI = [0–0.027], binomial proportion confidence interval). Taken together, these data suggest high reliability and accuracy of the respiratory trigger circuit as well as the quantification of respiratory rate.

### Confirmation of heart signal presence in PZT with ECG

Within the PZT signal, there were noticeable smaller fluctuations between the respiratory peaks, which were most prominent when the sensor was placed underneath the chest of the animal (Fig 2A, *top*). We tested whether these smaller fluctuations originate from the motion of the chest wall attributable to cardiac contractions by simultaneously recording the electrical

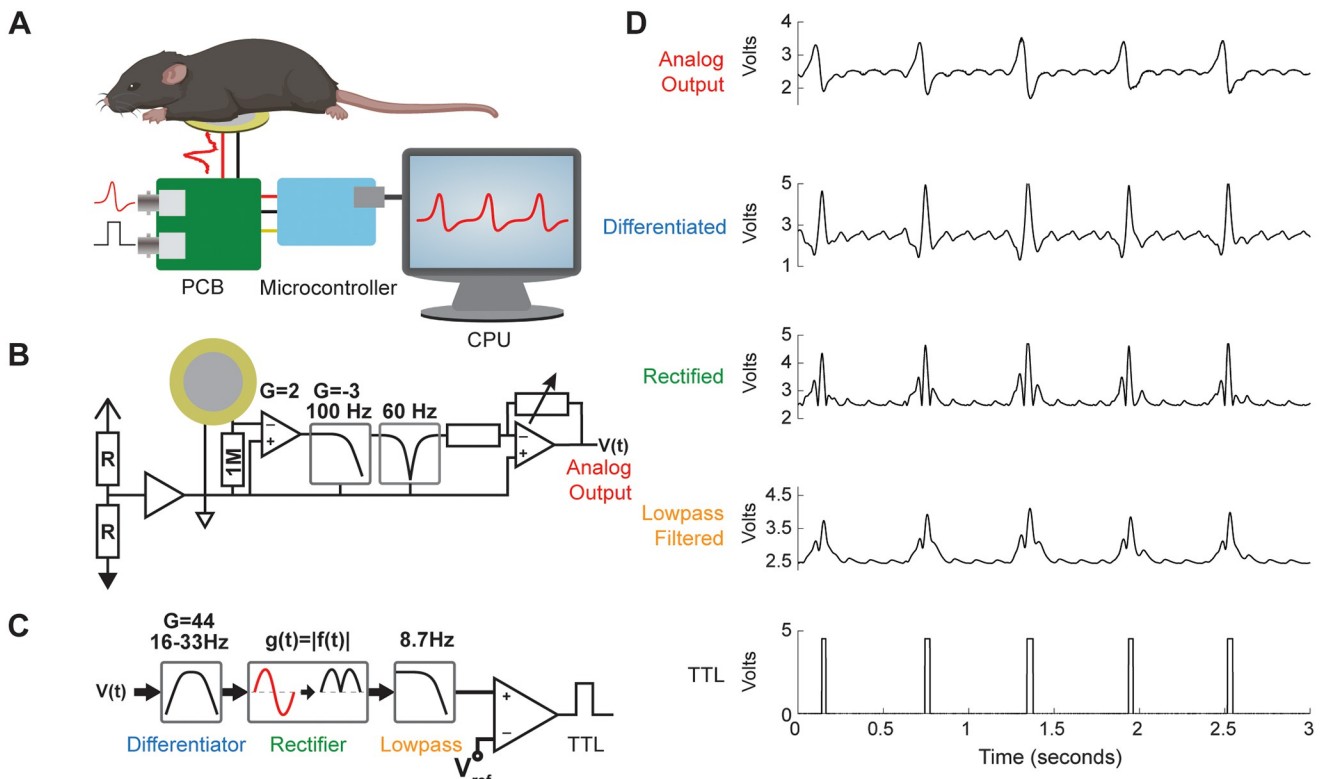

**Fig 1. MousePZT vital sign monitoring system for mice.** (A) Experimental setup for the system. The mouse rests on a piezoelectric sensor and the signal is amplified and filtered by a custom circuit, recorded with a microcontroller, and transmitted to a computer. (B) Simplified schematic for the custom circuit. The piezoelectric signal is amplified and lowpass filtered at 100 Hz, notch filtered to reduce line noise (50 or 60 Hz), and variably amplified. (C) Schematic of respiratory peak detection circuit. Signal from circuit in (B) is differentiated, rectified, and lowpass filtered before being passed to a comparator with a variable threshold. (D) Examples of a typical signal at each stage of the circuitry, with colored labels corresponding to circuit locations in (B) and (C).

activity of the heart with a three-lead ECG (Fig 3A). The small bumps in the PZT signal were locked to and occurred shortly after the QRS complex from the ECG signal (Fig 3B, *top*). We calculated heart rate from the MousePZT by finding the times of all heart beats and respirations, then calculating the median heart rate while excluding peaks obscured by the respiratory waveforms (Fig 3B, *bottom*; see Methods).

To test the ability of the MousePZT to detect acute changes in heart rate, we administered epinephrine intramuscularly in isoflurane-anesthetized mice. The MousePZT-based measurements of heart rate were shown to closely follow the rates determined from ECG in mice where changes in heart rate were induced pharmacologically with intramuscular epinephrine administration (Fig 3C, n = 3).

Across multiple mice (n = 10) and recordings, the absolute error between the PZT- and the ECG-based measurement of heart rate was 3.4 ± 13 beats per minute (bpm), or a relative error of 0.7 ± 2.6% ($R^2$ = 0.84, p < 0.001, Pearson's correlation; Fig 3D, black circles). For comparison, measurements derived from a commercial pulse-oximetry system (MouseOx) produced an absolute error of -1.2 ± 13 bpm with relative errors of -0.25 ± 2.8% ($R^2$ = 0.80, p < 0.001, Pearson's correlation; Fig 3D, white squares). When examining the source of errors for the piezo measurements, we found that the largest deviations occurred when the ratio of the heart rate to the respiratory rate dropped below 4 (Fig 3E), where heartbeats would overlap with

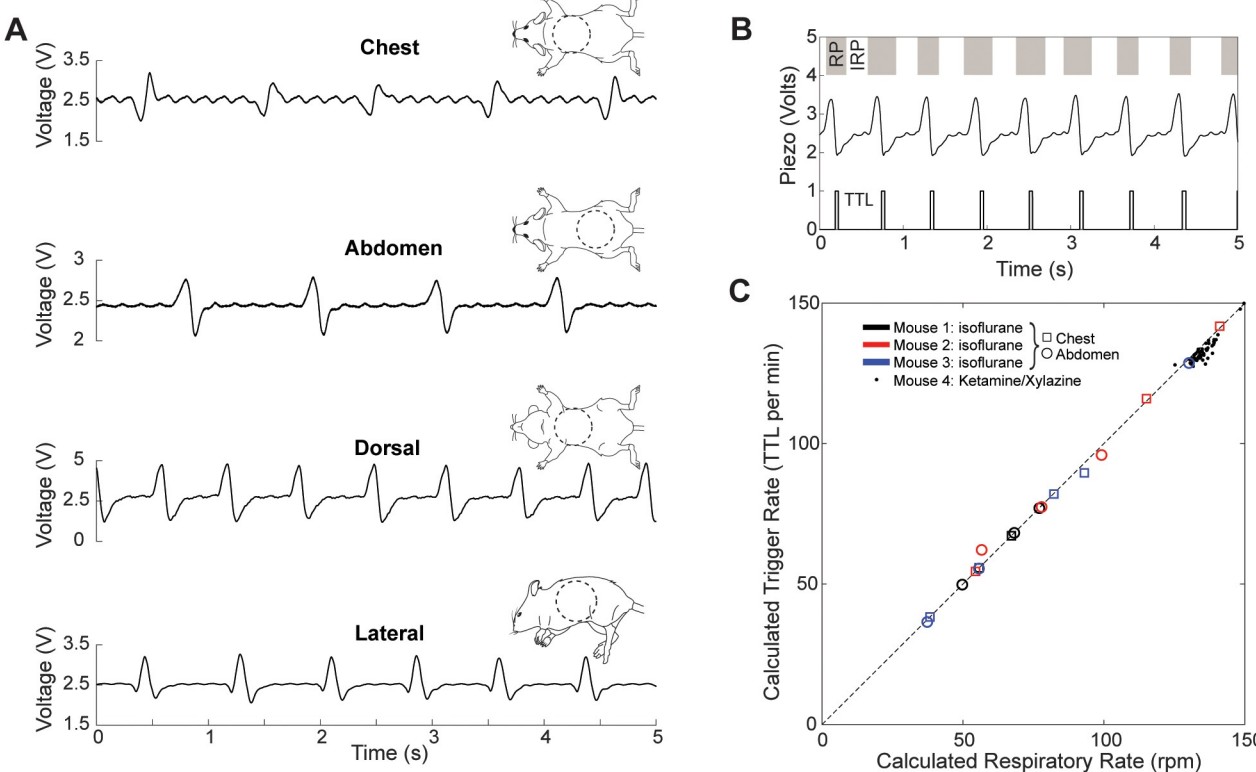

**Fig 2. MousePZT provides a robust and accurate measurement of respiratory rate as well as a respiration locked trigger signal. (A)** Examples of MousePZT signal with different sensor placement and animal orientation, as indicated in the schematics to the right of the traces (seen from above with sensor placement outlined by the dashed lines). **(B)** Example of the respiratory trigger. Respiratory phase (RP, gray) and inter-respiratory phases (IRP, white) are shown at top. Output piezo signal is shown in the middle, and the TTL respiratory trigger is at the bottom. **(C)** Correlation between breathing rates detected in 60-s intervals using the analog respiratory trigger circuit (Calculated Trigger Rate) and that from the MousePZT signal processed in MATLAB (Calculated Respiratory Rate) ($R^2 = 0.997$). Plot includes data from three mice anesthetized on isoflurane with both chest and abdominal sensor placements, as well as 60 60-s epochs from a mouse anesthetized on ketamine/xylazine with a chest sensor placement.

respiratory waveforms, often leaving less than two successive detectable beats between respirations.

## Dependence of heart rate and respiratory rate on anesthetic depth

Finally, we assessed the performance of the MousePZT under inhalant or ketamine/xylazine anesthesia and examined the correlation of the respiratory rate and heart rate to level of anesthetic for an animal. Under isoflurane in medical air, we randomly adjusted the isoflurane concentration from a starting value of 1.5%, and between 0.5 and 2.5%, always allowing time for the animals to stabilize (n = 7, one omitted due to data not being collected for multiple sensors; Fig 4A and 4B). Respiratory rate was strongly correlated to the inhaled isoflurane concentration ($R^2 = 0.81$, p < 0.001, Pearson's correlation; Fig 4C). Meanwhile, heart rate varied little with the inhaled isoflurane concentration ($R^2 = 0.043$, p = 0.23, Pearson's correlation; Fig 4D).

For ketamine and xylazine dependence, we administered a single dose, intraperitoneally, to six mice (one omitted due to insufficient warming) and tracked respiratory rate using the MousePZT and heart rate using ECG following induction and until recovery, as determined by return of mobility. Four of five mice included received supplemental oxygen after noting the drop in oxygen saturation (SpO2) previously described in the literature [25] and confirmed

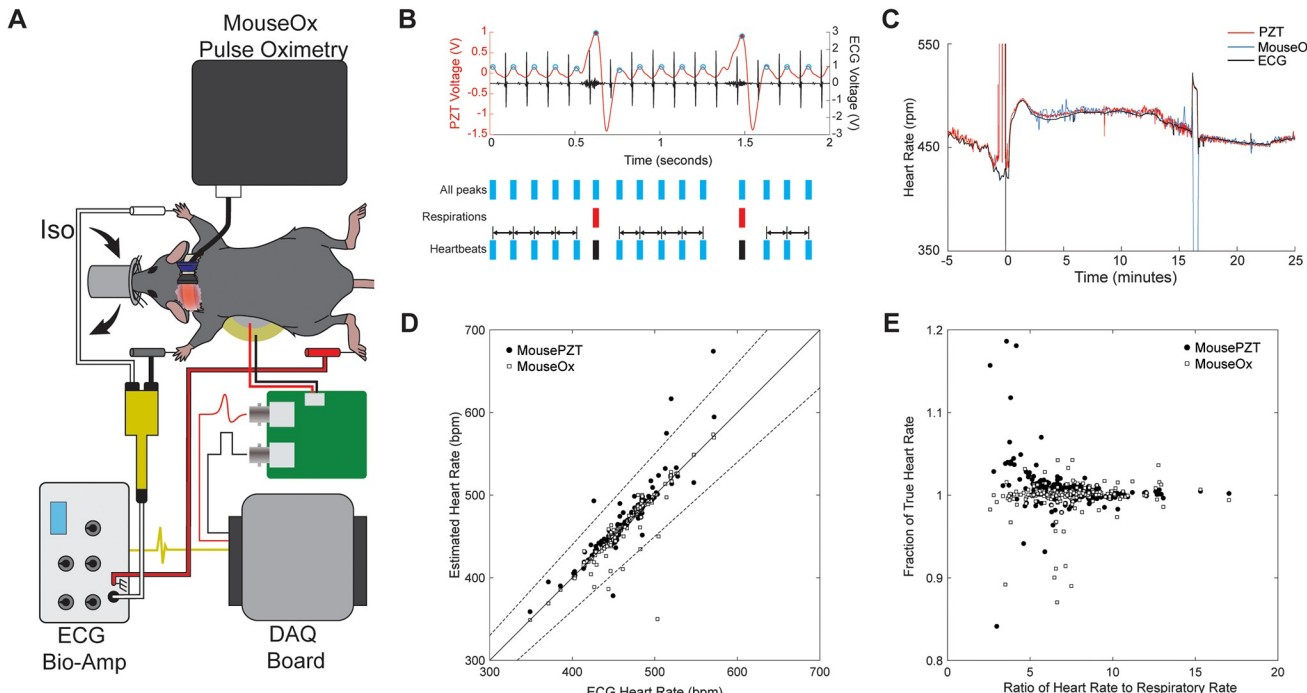

**Fig 3. Detection of heart rate with MousePZT. (A)** Schematic for simultaneous recording using the MousePZT, ECG, and MouseOx Pulse Oximeter. **(B)** Simultaneous recording of animal ECG (black) and MousePZT signal (red) (top). The QRS complex precedes the peak in the MousePZT signal. Schematic of the method for extracting heart rate by detecting all peaks, cardiac and respiratory, then removing respiratory peaks, and determining the inter heartbeat interval for the remaining cardiac peaks (bottom). **(C)** Example of heart rate changes in response to epinephrine (30–50 µL at 1 mg/mL, IM), as measured by MousePZT, ECG, and pulse-oximetry, simultaneously. All sensors show increases in heart rate following epinephrine administration. The MousePZT signal exhibits some motion artifact at the injection time. **(D)** Heart rate as measured by MousePZT (black circles) and pulse-oximetry (white squares) versus heart rate as measured using ECG. Dashed lines represent the boundary for 10% error. **(E)** Error in measuring heart rate using MousePZT (black circles) and pulse-oximetry (white squares) relative to measurements from ECG as a function of the ratio of heart rate to respiratory rate.

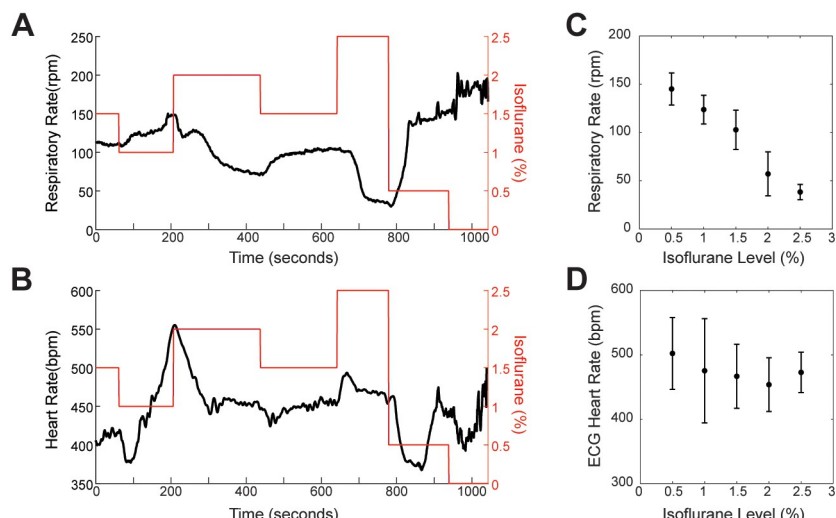

**Fig 4. Respiratory rate exhibits larger changes than heart rate in response to changes in inhaled isoflurane concentration.** Example measures of **(A)** respiratory rate and **(B)** heart rate (solid lines) as isoflurane concentration (dashed lines) are varied over time. Average **(C)** respiratory rate and **(D)** heart rate as a function of inhaled isoflurane concentration, with measurement taken from the last 30 s prior to changing concentration (n = 7 mice).

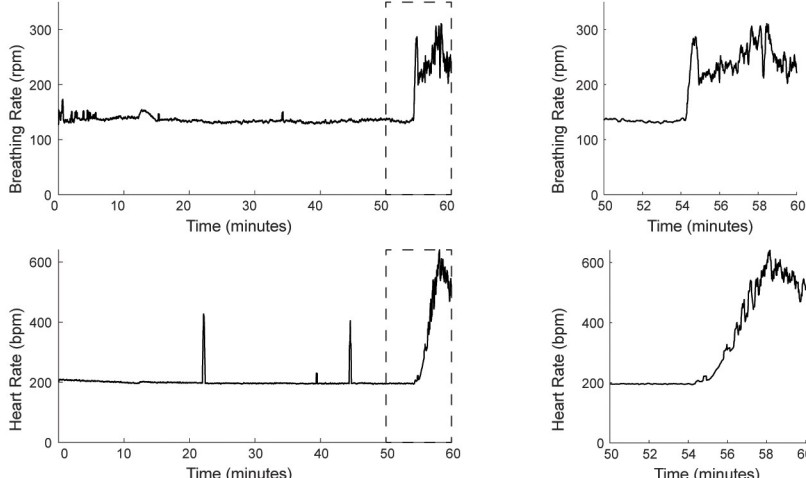

**Fig 5. Respiratory rate changes provide an early indicator of anesthetic depth changes.** Heart rate (top), as measured by ECG, and respiratory rate (bottom), as measured by MousePZT, following anesthesia with ketamine/xylazine and continuing until ambulatory recovery at ~60 min. Respiratory rate showed an earlier and faster increase than heart rate as seen in the final 10 min. prior to ambulation (right panels).

with pulse-oximetry. In all mice, there were increases in respiratory and heart rate that were evident a few minutes before the animals began to ambulate (Fig 5). The increase in the respiratory rate was detected, on average, 32 ± 16 s prior to any appreciable changes in the heart rate (p = 0.012, paired t-test; Fig 5). Taken together, these results show that careful monitoring of respiratory rate can provide an early and sensitive indicator of changes in anesthetic depth.

## Discussion

We have developed an open-source system that allows for quantification of respiratory rate and heart rate in anesthetized mice using a simple piezoelectric sensor. Additional circuitry produces TTL pulses for respiratory gating. The device can be fabricated in-house at low cost. Signals can be obtained without any additional preparation of the animal, such as hair removal or electrode insertion as needed by other methods. The determined respiratory and heart rates were shown to closely match those of other methods and a commercial system. In addition, the respiratory trigger signal was found to be highly accurate with minimal tuning required.

With our system, the MousePZT produced distinguishable and repeatable signal waveforms for respiratory rate measurements. Gomes et al. similarly used piezoelectric sensors placed under neath the mice, and noted that signals can be clearly identified with the animal in both ventral and dorsal recumbency [19]. We observed similar signal clarity and additionally were able to detect a pronounced respiratory signal with the animal in lateral recumbency (Fig 2A) as well as in minimal contact conditions under a limb, and at the base of the tail. The latter two locations enable placement of the device far from a site at which an intervention may be performed. However, the heartbeat waveform was attenuated (Fig 2A) or entirely absent with the sensor placed farther from the chest. For some animals and sensor placements, the waveform was less pronounced or differed in shape. Common causes of reduced respiratory signal clarity were from displacement of the sensor over time, or use of a soft base material under the animal. Signal clarity was quickly restored by adjustment of the animal or sensor position.

Previously developed low-cost approaches for determination of respiratory rate have comparable reliability to our MousePZT system. The light-emitting diode (LED)/photodiode

combination used by Lemieux and Glover [15] gives very strong respiratory signals from the change in reflectivity due to the motion of the chest wall, providing both a means to measure respiratory rate and produce a respiratory trigger using a software comparator. A thermistor implanted in the nasal cavity, used by McAfee et al. [18], produces more reliable signals than noninvasive temperature probes, but requires a moderately invasive surgical procedure. Without nasal epithelial implantation, thermistors can saturate following large breaths or sighs, leading to missed respirations [24].

Piezoelectric signals have been used previously to monitor respiratory rate but have lacked the capability for a respiratory trigger. The output TTL of the analog peak-detection circuit was shown to be very reliable and could be used for respiratory gating in applications such as CT or MRI to reduce respiratory motion artifacts. MRI-compatible piezoelectric materials are available and can be modified for gating applications [26]. Due to the sensitivity of the piezo sensor, the sensor can be placed around the animal to not interfere with imaging specific regions. The sensitivity of the detection circuit can be adjusted with a potentiometer to capture more or less of the respiratory waveform, such as detecting the onset of inspiration or the transition from inspiration to expiration, respectively. In experiments described here, the threshold was set manually but could be set, adaptively, by using a lowpass filtered (cutoff frequency < 0.1 Hz) modified Pan-Tompkins signal as the negative input to the comparator. While sensitivity can be increased to pick up some heartbeats (Fig 1D, lowpass filtered ripples), the hardware cannot pick up or distinguish between heartbeats and respirations. In addition, residual sensor noise may also be picked up intermittently with heartbeat detection. Therefore, in this paper, we focus on the use of the circuit for respiratory, but not cardiac, gating.

Heart beats also produce signals that can be picked up using a piezoelectric sensor, although they are more difficult to pick out than respiratory signals. Adding the capability to measure heart rate, without additional experimental or sensor complexity, provides another important vital sign to measure and record. With the sensor underneath the chest or abdomen and the mouse in dorsal recumbency, clear peaks from heart beats could be detected. Our algorithm that determines the time between successive heart beats, while identifying and ignoring regions where respiration obscures heartbeat signals, proved to provide a robust measurement of heart rate. Sato et al. [24] took a different signal processing approach to measure heart rate from piezoelectric sensors by applying a filter between 300 Hz and 1 kHz to capture the "heart sounds." We replicated this digitally, on removing the lowpass filter, but found it to be less reliable than our approach. "Heart sounds" were often obscured by respiratory vibrations, and, at lower heart rates, the time between the heart valves closing to produce the sounds was increased, leading to irregular heartbeat detection. The MousePZT system matched the accuracy of more invasive ECG and pulse oximetry-based heart rate measurements, as long as at least two cardiac cycles occur in the interval between respiratory cycles. This occurs when the heart rate is at least 3–4 times higher than the respiratory rate (Fig 3E) in which at least two successive heart beats will occur without being masked by respiratory peaks. This is a fundamental limitation based on the signal properties. In future work, additional methods, such as the use of wavelets or additional processing methods, may be able to better separate the respiratory and heartbeat signal waveforms. For this reason, heart rate was not measurable using MousePZT under ketamine/xylazine anesthesia, as heart rate (~200 bpm) was only 1.5 times higher than respiration rate (~135 rpm), which is a typical range for this anesthetic. Under isoflurane, heart rate is much faster, while respiration rates are similar or slower to ketamine/xylazine anesthesia, depending on isoflurane concentration and anesthetic depth [2] in these anesthetics, allowing for more accurate measurements of heart rate.

We showed that respiration rate is strongly correlated to isoflurane concentration and found that the respiratory rate increased sooner and faster than heart rate in animals

recovering from ketamine/xylazine anesthesia. Other studies have similarly demonstrated that respiration rate is heavily influenced by anesthetic depth [1, 2, 12, 19], making this vital sign a robust metric to monitor. This does not mean that heart rate should be ignored as altered heart rate can indicate other conditions such as hypothermia or responses to stressful stimuli [1, 12] that would otherwise be missed with respiratory signals alone.

In conclusion, we have demonstrated that a simple, low-cost sensor can be utilized for monitoring multiple vital signs in anesthetized mice with high accuracy across a range of sensor positions, animal orientations, and anesthetics used. MousePZT accurately measured respiratory rate and performed as well in determining the heart rate as other, more complex, monitoring methods, such as pulse-oximetry and ECG. Through the application of a simple peak-detection circuit, we further demonstrated high sensitivity and specificity of a respiratory-locked trigger signal. MousePZT is an effective tool for vital sign monitoring in anesthetized mice and is particularly useful when experimental conditions (e.g., imaging under low light) make direct visualization of the animal difficult. All resources to enable construction of a MousePZT, including a detailed parts list, a step-by-step construction guide for assembly of the circuit and overall system, ready-to-use design files for the printed circuit boards and the 3D printed housing, all microcontroller code and computer software, as well as a troubleshooting and debugging guide, are available at github.com/sn-lab/Mouse-Breathing-Sensor.

## Materials and methods

### MousePZT system

**Sensor circuit.** An Arduino microcontroller serves as the data acquisition board and 5-V power supply for the circuit. A virtual ground is created with two 10-kOhm resistors and a single-supply op-amp as a buffer. The sensor is a 27-mm diameter piezoceramic bender (PUI Audio, Inc.) with two soldered leads and electrical tape as insulation.

The spectrogram of several PZT recordings revealed prevalent peaks at the line noise with additional odd harmonics $3f$, $5f$, $7f$, etc. As many signal frequencies are expected to be harmonics of respiratory rate ($f \leq 3$ Hz) and heart rate ($f \leq 10$ Hz), a notch filter was applied for the initial line noise frequency, and a lowpass filter was used for suppressing subsequent harmonics while minimizing further attenuation or phase delays of the slower physiological signals. For the resulting circuit, the PZT is tied to the virtual ground via a 1-MOhm resistor to produce a highpass filter with the intrinsic capacitance of the PZT. The signal from the PZT is amplified through a noninverting amplifier (gain = 3 [9.5dB]) before passing through a secondary inverting amplifier and low-pass filter (gain = 2 [6 dB], $f_c$ = 100 Hz), followed by an adjustable twin-t notch filter centered on the line noise frequency (on-board potentiometers tune this for 50-Hz or 60-Hz line noise). The signal is finally passed through a noninverting amplifier with a variable gain set by a potentiometer and read out by the on-board analog-to-digital converter of the microcontroller. A simplified experimental setup and schematic of the MousePZT circuit are depicted in Fig 1A and 1B, respectively.

**PZT-derived respiratory rate and heart rate calculation.** The microcontroller communicates with an app written in MATLAB's App Designer via a serial port and transmits the signal voltage at a sampling rate specified by the app. The baud rate is set to 115,200 bits/s to balance speed and stability, limiting the max theoretical sampling frequency ($f_s$) due to communication to approximately 7.2 kHz. As the signals of interest are low frequency (HR < 10 Hz), the signal data is boxcar filtered with the window size ($w_s$) set to remove the line noise and resulting harmonics completely ($w_s = f_s$/[line frequency]). Once per second, MATLAB takes a 5-s window of the boxcar-filtered data and locates the respiratory peaks using find-peaks(). Detected respiratory peak intervals are averaged to obtain the respiration rate for that

5-s window. For determination of heart rate from the PZT signal, respiratory and heart rate peaks are detected from the digitally filtered signal without additional processing using find-peaks(), with the peak prominence and minimal distance between peaks settings significantly reduced to enable detection of both heart beat associated peaks and respiratory peaks. Detected peak locations that overlap with the initial respiratory peaks are converted to NaNs as the respiratory signal dominates the heart signal in those intervals. The heart rate is then calculated from the median of the time interval between successive heart beats, with NaN intervals omitted (Fig 3A).

**Analog respiratory trigger circuit.** For analog detection of respiratory peaks, a modified Pan-Tompkins analog circuit [27] (Fig 1C) was implemented in which the signal is differentiated, rectified (not squared), and lowpass filtered. To avoid saturation, the output respiratory signal of the sensor circuit is reduced to 1/18 of the original amplitude using a voltage divider to the virtual ground and buffered before being differentiated with corner frequencies of 16 Hz and 33 Hz and a resulting gain of 44 [33 dB]. The differentiated signal is sent through a full-wave rectifier before being low passed at 8.7 Hz. A rectifier was opted for rather than a squarer circuit to reduce circuit complexity and avoid exceeding the voltage range available. A comparator outputs a TTL pulse when the processed signal exceeds a reference voltage set by a potentiometer. The voltage signals at each stage of this circuit are depicted in Fig 1D. Before each recording with the analog peak detection, the two potentiometers in the circuit were used to adjust the signal amplitude and trigger sensitivity. The potentiometer for the output signal was adjusted to produce a 1–2 V peak-to-peak amplitude due to respiration, as viewed in the MATLAB app or on an oscilloscope. The second potentiometer for the respiratory peak-detection circuit was adjusted to avoid erratic triggering (an on-board LED flashes with each trigger to facilitate easy tuning) and to either lower the threshold to enable triggering across the entire respiratory waveform or increasing the threshold to trigger just at the transition from inspiration to expiration.

## In vivo validation studies

**Animals.** All animal procedures were approved by the Cornell Institutional Animal Care and Use Committee (protocol 2015–0029) and were performed under the guidance of the Cornell Center for Animal Resources and Education. We used 15 adult C57BL/6 mice of both sexes between 3 and 8 months of age, and between 19 and 32 grams. Some mice were used in multiple experiments with at least 4 weeks of recovery in between. All mice recovered from the procedures performed in this study. Experimenters were blinded to video scoring of respiratory cycles, but not to experiments, other data, or analysis.

**ECG placement.** Three electrode needles were attached to the left forelimb, right forelimb, and left hindlimb of anesthetized mice, either subcutaneously in mice receiving epinephrine, or noninvasively via electrode paste (SignaCreme, Parker Laboratories) for mice expecting to regain ambulation during recording. ECG traces were collected from Lead I, based on Einthoven's triangle configuration. The leads were fed into a bioamplifier (ISO-80, World Precision Instruments) with gain set to $10^3$ [60 dB] and the bandpass filter set to 0.1–300 Hz.

**Pulse-oximetry.** Some mice were recorded with an additional commercial pulse-oximeter (MouseOx, StarrLife Sciences) to measure heart rate, respiratory rate, and additionally track SpO2, either using the manufacturer's thigh sensor (n = 3 mice, 3 recordings) or throat collar (n = 7 mice, 12 recordings). For placement, the hair on the thigh (for mice receiving epinephrine injections) or neck (for mice undergoing isoflurane variation or ketamine/xylazine administration) was shaved and followed by treatment with depilatory cream (Nair). Vitals

were recorded at either 1, 5, or 15 Hz with files timestamped for synchronization to ECG and PZT acquisition.

**Isoflurane anesthesia.** Anesthesia was induced with 3% isoflurane in medical air using an induction chamber. Once unresponsive to toe or tail pinch, the animal was moved to a stereotactic frame with a nose cone and a feedback-controlled heating pad (40-90-8D, FHC) set to 37˚C. Puralube vet ointment was applied to each eye and reapplied as necessary while the animal was anesthetized. Mice were given a subcutaneous injection of glycopyrrolate (0.2 mg/kg, 20 mg/mL, West-Ward Pharmaceuticals). Isoflurane was reduced to 1–2% for maintenance during sessions involving epinephrine administration.

For assessing the effect of isoflurane on heart rate and respiratory rate, isoflurane was held at 1.5% while the animal was prepped with noninvasive ECG leads and the pulse-oximeter collar for the experiments. After 10 minutes following the onset of the recording session, isoflurane concentration was varied randomly between 0.5–2.5% in 0.5% intervals and maintained until the vitals of the animal stabilized or the SpO2, as measured by the pulse-oximeter, dropped below 85% after which the isoflurane was reduced. Measurements for comparison of heart rate and respiratory rate were taken from the last 30 s prior to adjusting anesthesia to the next concentration.

**Setting and verifying the respiratory peak detector.** To determine the accuracy of the respiratory trigger, the output PZT signal and peak detector output TTL signals were recorded for 60-s epochs on a USB-6003 DAQ board (National Instruments). The modified Pan-Tompkins algorithm was applied to the respiratory signal and the respiratory peaks and peak widths were determined using findpeaks() in MATLAB. The signal was split into respiratory and inter-respiratory epochs based on the respiratory peak locations and respective widths. Each epoch was then examined for high or low TTL values to determine the true positive (TTL during respiration), true negative (no TTL outside respiration), false positive (TTL outside respiration), and false negative (no TTL during respiration) rates. For calculation of respiratory rate using the trigger signal, we applied a triangular moving average filter on the TTL signals before differentiating and running the findpeaks() function to identify the rising pulse edges for calculation of the TTL-based respiratory rate. Alternatively, for visual accuracy of the peak-detector, a LED was tied to the output of the comparator to indicate when the output trigger was high, and the mouse and LED were recorded at ~30 fps using a cell phone camera. The video was stabilized in FIJI (ImageJ) using the Linear Alignment with SIFT plugin and cropped so that reviewers could score the start and stop of respiratory cycles without input bias from the LED. A region-of-interest (ROI) was placed over the LED in the stabilized video to determine, based on pixel intensity, when the trigger was active.

**Pharmacological manipulation of heart rate.** Three mice were anesthetized with isoflurane anesthesia as described above. MousePZT and ECG signals were recorded at 10 kHz on the DAQ board and the calculated pulse oximeter vital signs at 15 Hz using MouseOx software. Mice were recorded for up to 5 minutes to achieve a baseline rate, followed by intramuscular administration of 30–50 µL epinephrine (1 mg/mL) into the right hindlimb to induce a notable increase in heart rate. Mice were recorded for at least 30 minutes following epinephrine injection.

**Ketamine/xylazine administration.** Five mice were given a single dose of ketamine/ xylazine solution (0.1 mL/10 g body weight; 10 mg/mL ketamine and 1 mg/mL xylazine) intraperitoneally. Once unresponsive to toe or tail pinch, the mice were placed in ventral recumbency on a heating pad with temperature feedback set to 37˚C. Puralube vet ointment was applied to each eye and reapplied as necessary while the animal was immobile, and mice were injected with glycopyrrolate as described above. The PZT sensor was then placed between the thorax and the heating pad. ECG leads were noninvasively connected using

electrode paste. ECG and PZT signals were recorded on the USB-6003 DAQ board at 10 kHz or on the USB-6001 DAQ board at 1 kHz or 5 kHz until the animal showed signs of ambulation. Four of the mice received supplemental oxygen with pulse-oximeter throat collars placed as described above.

**ECG-derived heart rate calculation.** Heart rate was determined by detection of the QRS complexes in the ECG signal [28]. Typically, these peaks have a large single peak and therefore can be detected simply using MATLAB's findpeaks() function. The peak-to-peak interval is then averaged across 5-s windows to produce a heart rate measurement that is compared to that from the PZT sensor. In cases where the *J*-wave nearly matches or exceeds the *R*-wave amplitude, resulting in erroneous heart rate determination [28], we apply the Pan-Tompkins algorithm with corner frequencies at 30 and 100 Hz, and a boxcar filter of 10 ms for the squared derivative.

## Statistical methods

All statistical tests were performed in MATLAB using the Statistics and Machine Learning Toolbox. We used a Pearson's correlation test for determining the level of agreement between measures of respiratory rate (Fig 2C) and heart rate (Fig 3D) between sensor methods and calculations as well as the significance of the trend between isoflurane concentrations and either respiratory rate (Fig 4A and 4C) or heart rate (Fig 4B and 4D). When comparing errors in measuring heart rate, absolute error was defined as the difference between the measured rate from the device being tested (MousePZT or pulse-oximetry) to the ground truth measured by ECG while relative error was defined as the absolute error as a percent of the ground truth measurement. For visual inspection of the respiratory peak detector, we used a binomial proportion confidence interval to give the probability that the peaks were detected correctly using the manual scoring of the videos as ground truth (Fig 2B). The times at which significant changes in respiratory and heart rates following ketamine/xylazine anesthesia were detected prior to recovery were determined using the findchangepts() function in MATLAB. A paired Student's t-test compared the times of change in respiratory rate to the times of change in heart rate respective to each animal (Fig 5).

## Supporting information

**S1 Movie. Real-time respiratory detection example.**
(AVI)

## Acknowledgments

We thank Jordan Harrod, Riona Reeves, Rohan Roy, Julia Telischi, and Kelly Wilson for their early contributions in the design of this device.

## Author Contributions

**Conceptualization:** Daniel A. Rivera, Chris B. Schaffer.

**Data curation:** Daniel A. Rivera.

**Formal analysis:** Daniel A. Rivera.

**Funding acquisition:** Chris B. Schaffer.

**Investigation:** Daniel A. Rivera, Anne E. Buglione, Sadie E. Ray.

**Methodology:** Daniel A. Rivera, Anne E. Buglione.

**Project administration:** Chris B. Schaffer.

**Resources:** Daniel A. Rivera.

**Software:** Daniel A. Rivera.

**Supervision:** Chris B. Schaffer.

**Validation:** Daniel A. Rivera, Anne E. Buglione.

**Visualization:** Daniel A. Rivera.

**Writing – original draft:** Daniel A. Rivera.

**Writing – review & editing:** Daniel A. Rivera, Anne E. Buglione, Chris B. Schaffer.

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
