## [Decision Letter · Decision Letter 0]

4 Dec 2023

PONE-D-23-27450MousePZT: a simple, reliable, low-cost device for vital sign monitoring and respiratory gating in mice under anesthesiaPLOS ONE

Dear Dr. Schaffer,

Thank you for submitting your manuscript to PLOS ONE. After careful consideration, we feel that it has merit but does not fully meet PLOS ONE’s publication criteria as it currently stands. Therefore, we invite you to submit a revised version of the manuscript that addresses the points raised during the review process.

We look forward to receiving your revised manuscript.

Kind regards,

Li-Ming Hsu

Academic Editor

PLOS ONE

Journal Requirements:

This work was supported by the National Institute on Aging (https://www.nia.nih.gov/), grant number, AG049952 (CBS). The funders had no role in study design, data collection and analysis, decision to publish, or preparation of the manuscript.

Reviewers' comments:

Reviewer's Responses to Questions

**Comments to the Author**

1. Is the manuscript technically sound, and do the data support the conclusions?

Reviewer #1: Yes

Reviewer #2: Yes

2. Has the statistical analysis been performed appropriately and rigorously? 

Reviewer #1: I Don't Know

Reviewer #2: I Don't Know

3. Have the authors made all data underlying the findings in their manuscript fully available?

Reviewer #1: Yes

Reviewer #2: Yes

4. Is the manuscript presented in an intelligible fashion and written in standard English?

Reviewer #1: Yes

Reviewer #2: Yes

5. Review Comments to the Author

Reviewer #1: General comments:

The authors presented a well written and organized manuscript reporting development of a low cost, feasibe PZT device to measure vital signs of the cardiovascular and respiratory parameters. The methodology was used in a sound manner and the results are informative. The discussion was constructive with rationale behind the variable factors underlying the physiological changes. Thus, the paper deemed acceptable for publication. Minor updates may be completed.

Specific comments :

Introduction

Missing letters in words:

-Placing

-Pattern

Methods

-Please specify the number of mice underwent pulse oximetry

Figures

-Figure 1a: It'd be advisable to support the methods section with a fully annotated diagram or real picture demonstrating the set up of ECG, pulse oximetry and respiratory PTZ devices in the mice.

Thank you.

Reviewer #2: 1.The article didn't mention the specific statistical methods for some comparison.

2.Further refinement of MousePZT could involve extensive calibration and validation under various physiological conditions to improve its accuracy and reliability, how to figure out the identified issue related to heart rate to respiratory rate ratios below 4.

3.How to remove and define the signal noise during signal analysis?

Overall, the work with MousePZT represents a significant contribution to the field by offering an effective, accessible, and accurate tool for vital sign monitoring in anesthetized mice, fostering progress in scientific research and experimentation.

6. PLOS authors have the option to publish the peer review history of their article (what does this mean?). If published, this will include your full peer review and any attached files.

Reviewer #1: No

Reviewer #2: No

---

## [Author Response · Author response to Decision Letter 0]

23 Jan 2024

We thank the reviewers for their feedback regarding this manuscript. We have made attempts to address each of the reviewers’ comments in this response and within the manuscript.

Reviewer #1: General comments:

The authors presented a well written and organized manuscript reporting development of a low cost, feasible PZT device to measure vital signs of the cardiovascular and respiratory parameters. The methodology was used in a sound manner and the results are informative. The discussion was constructive with rationale behind the variable factors underlying the physiological changes. Thus, the paper deemed acceptable for publication. Minor updates may be completed.

Specific comments :

Introduction

Missing letters in words:

-Placing

-Pattern

Thank you for catching these mistakes. We have made these corrections.

Methods

-Please specify the number of mice underwent pulse oximetry

We apologize for not including these numbers. The manuscript has been updated to include the number of mice that underwent pulse oximetry for each sensor type, as well as the total sessions recorded.

Figures

-Figure 1a: It'd be advisable to support the methods section with a fully annotated diagram or real picture demonstrating the set up of ECG, pulse oximetry and respiratory PTZ devices in the mice.

We thank the reviewer for this suggestion. We have added a panel to Figure 3 with a schematic for placement of all three sensors on the animal for heart rate measurement comparisons across devices.

Reviewer #2: 1.The article didn't mention the specific statistical methods for some comparison.

2.Further refinement of MousePZT could involve extensive calibration and validation under various physiological conditions to improve its accuracy and reliability, how to figure out the identified issue related to heart rate to respiratory rate ratios below 4.

3.How to remove and define the signal noise during signal analysis?

1. We apologize for not elaborating on the statistical methods used. The statistics section was expanded to explain what statistical tests were used for each comparison made and why.

2. While, yes, the system can be improved for better accuracy and reliability especially for heart rate determination, the primary focus of this manuscript was for measurement and gating of respirations. The ability to measure heart rate is more limited, but possible in some circumstances. The primary limitation comes from masking of the peaks in the signal originating from the heartbeat by the respiratory waveforms. If the respiratory rate is not too fast, multiple heartbeats can be detected in between subsequent respiratory events, allowing for accurate measurement of heart rate. If respiratory rate is too fast, however, there are too few successive heart beats that are not masked by the larger amplitude respiratory waveforms. In fact, we show this increase in error in heart rate measurement as function of the ratio of the heart rate to the respiratory rate in Fig. 3, where the error increases as this ratio decreases. As such, this is a fundamental limitation of the signal and therefore, measuring heart rate accurately will be case-dependent. Primarily, this depends on how a particular anesthetic agent influences heart and respiratory rates. For example, in the data in our manuscript, measuring heart rate under isoflurane is reliable while under ketamine/xylazine the heart rate, where heart rate is slower, the measurement is confounded. Additional signal processing methods for unmixing of the two waveforms would be outside the scope of this manuscript.

3. We believe this is a good point to include should others want to adjust the electronic filters used. In initially examining the power spectrum from the PZTs, we noticed peaks at the line noise frequency and odd harmonics. We expect most physiological signals, including harmonics, to be well below these frequencies. This reasoning for using the notch filter and lowpass filter has been added to the Methods regarding construction of the device.

Overall, the work with MousePZT represents a significant contribution to the field by offering an effective, accessible, and accurate tool for vital sign monitoring in anesthetized mice, fostering progress in scientific research and experimentation.

---

## [Editor Report · Decision Letter 1]

5 Feb 2024

MousePZT: a simple, reliable, low-cost device for vital sign monitoring and respiratory gating in mice under anesthesia

PONE-D-23-27450R1

Dear Dr. Schaffer,

We’re pleased to inform you that your manuscript has been judged scientifically suitable for publication and will be formally accepted for publication once it meets all outstanding technical requirements.

Kind regards,

Li-Ming Hsu

Academic Editor

PLOS ONE

---

## [Editor Report · Acceptance letter]

23 Feb 2024

PONE-D-23-27450R1 

PLOS ONE

Dear Dr. Schaffer, 

I'm pleased to inform you that your manuscript has been deemed suitable for publication in PLOS ONE. Congratulations! Your manuscript is now being handed over to our production team.

Kind regards, 

on behalf of

Dr. Li-Ming Hsu 

Academic Editor

PLOS ONE